# Detoxification of Fumonisins by Three Novel Transaminases with Diverse Enzymatic Characteristics Coupled with Carboxylesterase

**DOI:** 10.3390/foods12020416

**Published:** 2023-01-16

**Authors:** Yue Wang, Junhao Sun, Mengwei Zhang, Kungang Pan, Tianhui Liu, Tongcun Zhang, Xuegang Luo, Junqi Zhao, Zhongyuan Li

**Affiliations:** 1State Key Laboratory of Food Nutrition and Safety, College of Biotechnology, Tianjin University of Science and Technology, Tianjin 300457, China; 2School of Chemical and Biological Engineering, Qilu Institute of Technology, Jinan 250200, China

**Keywords:** fumonisin, transaminase, carboxylesterase

## Abstract

Fumonisin (FB) is one of the most common mycotoxins contaminating feed and food, causing severe public health threat to human and animals worldwide. Until now, only several transaminases were found to reduce FB toxicity, thus, more fumonisin detoxification transaminases with excellent catalytic properties required urgent exploration for complex application conditions. Herein, through gene mining and enzymatic characterization, three novel fumonisin detoxification transaminases—FumTSTA, FumUPTA, FumPHTA—were identified, sharing only 61–74% sequence identity with reported fumonisin detoxification transaminases. Moreover, the recombinant proteins shared diverse pH reaction ranges, good pH stability and thermostability, and the recombinant protein yields were also improved by condition optimum. Furthermore, the final products were analyzed by liquid chromatography-mass spectrometry. This study provides ideal candidates for fumonisin detoxification and meets diverse required demands in food and feed industries.

## 1. Introduction

As one of the most prevalent foodborne fungal toxins, fumonisins (FBs) are mostly generated by fungi (*Fusarium verticillioides* and *Fusarium oxysporum*) and widely distributed in animal feed and human food [1,2,3]. FBs have an extremely stable chemical structure consisting of two methyl groups (-CH_3_), one amino group (-NH_2_), one to four hydroxyl groups (-OH), and two tricarboxylate groups (TCA) in different positions as well as linear polyketide derived backbones [4]. Since they have a similar molecular structure to sphingosine and sphinganine, fumonisins exert significant toxicity by interfering sphingolipid metabolism and inducing oxidative stress [4,5,6].

To date, more than 15 fumonisin congeners have been found, and fumonisins B1 (FB1) is the most toxic and abundant fumonisin [7,8]. Based on many toxicological studies, FB1 has been demonstrated to induce a wide range of diseases in animals, such as cerebral leukodystrophy in horses, pulmonary edema syndrome in pigs, nephrotoxicity in rats, and apoptosis in many other cell types [9,10,11]. Moreover, epidemiological investigations have indicated that fumonisins are related to esophageal cancer, primary liver cancer, neural tube defects, and cardiovascular disease in highly exposed human populations [12,13]. Due to its significant risk to human health and animal welfare, many international organizations have already released the temporary maximum amount of fumonisins in food, feed, and feed raw materials [8,14]. Both social and scientific concerns have also focused on controlling and resolving fumonisins contamination [15].

Fumonisins are highly soluble in water and very thermostable, thus it is challenging to destroy and remove them from food and feed [7,15]. The current detoxification strategies of fumonisins contain physical, chemical, and biological detoxification methods. Physical methods including adsorption, heating and irradiation can detoxify FB in contaminated feedstocks [16,17,18]. For example, adsorption with silico-aluminate, montmorillonite, bentonite is very widely used in marketed fumonisins adsorbent. However, adsorption with adsorbents cannot completely remove fumonisins from grains, and some nutrient substances were also adsorbed at the same time [16,19]. As for chemical methods, FB chemical structure can be destroyed by strong alkalis or oxidants [17]. However, some chemical strategies such as alkali cooking and altering the structure of fungal toxins with acids, bases and oxidants may destroy nutrients and generate potentially toxic compounds [17,20].

As a safe, efficient, and environmentally friendly detoxification technology, biological detoxification was used to effectively detoxify fumonisins and has attracted more and more attention from researchers. For example, some microorganisms including lactic acid bacteria, *Phingomonas*, *Bacillus*, *yeasts*, *Klebsiella*, and other microorganisms, showed inhibitory effects on fumonisin-producing strains or adsorption and detoxication of fumonisins [7]. Moreover, nanobody Nb5 was found to have the effect of antagonizing FB1, which could reduce FB1 toxicity on embryonated eggs, improved body weight gain and reduced gizzard ulceration in broilers [19]. Recently, utilizing biological enzymes to convert mycotoxins into low or non-toxic derivatives has been recognized and studied for its high efficiency and safety [16,21,22]. Previous studies have shown that two TCA and -NH_2_ functional groups are recognized as the toxic moieties of fumonisins [23]. Consequently, the biodegradation of fumonisins by enzyme mainly focuses on the removal or replacement of two specific chemical structural components (TCA and -NH_2_ functional groups) by two steps [24]. In the case of FB1, the tricarboxylic acid groups are firstly cleaved by carboxylesterase to form hydrolyzed FB1 (HFB1). Then, HFB1 further reacts with transaminase enzyme or oxidative deaminase to form 2-keto-HFB1 as well as N-acetyl HFB1 or 2-OP1 semi-Ketal [25,26,27]. Previous studies found that these products were shown with less toxic compared with that FB1 [7].

Obtaining effective detoxification enzymes has important implications for promoting the biodegradation of fumonisin. Unfortunately, to the best of our knowledge, the potential transaminases used for HFB1 biodegradation are very limited (Table 1). Only transaminases named FumIS from *Sphingopyxis* sp. MTA144 and FumIB from bacterium ATCC 55552 were found to degrade HFB1 [26,28]. Hence, the gene mining and biochemical characterization of fumonisin detoxification enzymes need to be further explored. In this study, three potential fumonisin detoxification transaminases were identified based on protein sequence and structure analysis, and biochemical characterization showed their efficient fumonisin detoxification ability with diverse properties, which provides enzyme resource and solving strategy for fumonisins elimination.

## 2. Materials and Methods

### 2.1. Strains, Plasmids, Chemicals, and Enzyme

The plasmid pET-28a from Invitrogen (Carlsbad, CA, USA) was adopted as the expression vector in *Escherichia coli* DH5α and BL21 (DE3) were utilized to clone and expression of hosts, separately. The recombinant strains of *E. coli* were cultivated in LB medium (0.5% *w/v* yeast extract, 1% *w/v* tryptone, and 1% *w/v* NaCl). Phusion DNA polymerase, restriction endonuclease, and T4 DNA ligase were purchased from ThermoFisher Scientific (Shanghai, China). The plasmid extraction kit, PCR product purification kit, and isopropyl-β-D-thioga-lactopyranoside (IPTG) were purchased from Solarbio (Beijing, China). Nickel columns and nickel resin used for affinity chromatography were purchased from GE Health-care (Uppsala, Sweden). Substrate FB1 (purity of 98%) was obtained from Pribolab Ltd. (Qingdao, China, http://wwwpribolab.com/, accessed on 4 March 2022) and dissolved in acetonitrile-water. All additional reagents and chemicals are of analytical grade.

### 2.2. Gene Mining Novel Transaminases for Fumonisin Degradation

The potential gene pool of transaminases for fumonisin degradation was firstly established by BLASTP using transaminases from bacterium ATCC 55552 and *Sphingopyxis* sp. MTA144 as input models (https://blast.ncbi.nlm.nih.gov/Blast.cgi/, accessed on 7 March 2022). SignalP (version 5.0) server for forecasting signal peptide sequences (www.cbs.dtu.dk/services/SignalP/, accessed on 7 March 2022). Multiprotein sequence comparisons were conducted with ClustalW (http://www.ebi.ac.uk/clustalW/, accessed on 10 March 2022) and illustrated using the ESPript 3.0 online server (http://espript.ibcp.fr/ESPript/ESPript/, accessed on 10 March 2022). The MEGA6.0 program was used to construct a phylogenetic tree by repeating bootstrap 1000 times according to the neighbor-joining algorithm. The display of the phylogenetic tree was further refined with iTOL software (http://itol.embl.de/, accessed on 10 March 2022). The transaminase structures were predicted by AlphaFold2 (https://github.com/deepmind/alphafoldRoseTTAFold, accessed on 10 April 2022). Protein structure and catalytic sites were profiled by PyMOL software (https://pymol.org/2/, accessed on 10 April 2022). In order to evaluate their structural similarity, root mean square deviation (RMSD) was calculated.

### 2.3. Protein Expression and Purification of Transaminases

The gene sequences of transaminase FumUPTA, FumTSTA, and FumPHTA selected from the screening pool were codon optimized, gene synthesized, and cloned into the expression vector pET-28a(+) (GENEWIZ, Suzhou, China). The recombinant plasmids transformed into *E. coli* BL21(DE3) competent cells. Thereafter, the recombinant *E. coli* BL21(DE3) strains were incubated in LB medium containing 100 μg/mL ampicillin at 37 °C. Then, the target proteins were induced by IPTG at a final concentration of 1 mM for 20 h at 25 °C. Moreover, carboxylesterase FumDSB used for degrading the TCA groups of FB1 was induced at its optimal induced condition (30 °C, 0.6 mM IPTG, 20 h) [29].

To obtain the recombinant proteins of carboxylesterase FumDSB, transaminase FumUPTA, FumTSTA, and FumPHTA, the recombinant proteins were purified by nickel affinity chromatography (GE Health-care, Uppsala, Sweden). The molecular mass of the recombinant proteins was assayed by sodium dodecyl sulfate polyacrylamide gel electrophoresis (SDS-PAGE). The concentration of purified protein was determined by the Bradford method, using a protein assay kits (Bio-Rad, Hercules, CA, USA).

For further identification of the specific proteins, the purified proteins were detected by western blotting analysis based on their terminal His-tags [9]. After separation by polyacrylamide gel electrophoresis, the proteins were transferred to NC membranes (Beyotime Biotech). The membranes were sealed at room temperature for 1 h and hatched with primary antibody: 6-HIS (Anorun, Beijing, China) overnight at 4 °C. Subsequently, the goat anti-mouse IgG was labeled as secondary antibody using HRP (1:2000, Zhennuo Biotech, Shanghai, China). The staining protein was shown by scanning the membrane with Odyssey Infrared Laser Imaging System (LI-COR, Lincoln, NE, USA).

In addition, to optimize the expression level of recombinant protein, the protein expression yield of transaminases with compared at different incubation times IPTG concentrations, and induction temperatures.

### 2.4. Fumonisin Degradation by Transaminases

In this study, HFB1 was firstly obtained from the reaction mixture of FB1 and esterase FumDSB, 100 µL of FB1 solution (50 µg/mL) and 900 µL of purified enzyme FumDSB (1.164 mg/mL) were reacted at 37 °C for 2 h. The producing HFB1 were further identified by LC-MS, then they were used as the substrate for transaminases directly without any extraction and purification steps. For transaminase, each assay solution (100 µL) contained 95 µL of purified transaminase (1.38 mg/mL) in PBS buffer (8 g/L NaCl, 3.58 g/L Na_2_HPO_4_·12H_2_O, 0.2 g/L KCl, 0.27 g/L KH_2_PO_4_), 5 µL of HFB1 solution, and pyruvate solution, mixed to a final concentration of HFB1 of 1 μg/mL. The reactions were incubated at 37 °C for 12 h and then inactivated at 100 °C for 10 min.

The intermediate degraded products (HFB1) by transaminases were detected by HPLC. The chromatography was performed on an Agilent 1200 LC system (Agilent Technologies, Santa Clara, CA, USA). The samples were derivatized by adding 500 μL of o-phthaldialdehyde (OPA) derivative solution and 400 μL of aqueous acetonitrile, after gentle mixing, then immediately filtered through a 0.22 μm nylon filter and detected by LC-LFD within 5 min. The excitation wavelength (Ex) of the fluorescence detector was set at 335 nm, and the emission wavelength was set at 440 nm; the mobile phase (methanol: 0.1 mol/L NaH_2_PO_4_ = 77:23, *v*/*v*, adjust pH to 3.8 with phosphoric acid) was maintained at a flow rate of 1 mL/min. The temperature of the column was adjusted to 25 °C and the volume of injection was adjusted to 20 μL. The degradation rate of HFB1 = (original content of HFB1 − remaining content of HFB1)/original content of HFB1 × 100%. All experiments were duplicated thrice.

### 2.5. Analysis of Final Degradation Products by LC-MS

To identify the detoxification final product of FB1 by carboxylesterase and transaminases, the analysis of degradation products was performed on an LC-QTOF-MS system (Agilent Technologies, Santa Clara, CA, USA). The chromatographic column was ZORBAX Extend-C18 reversed-phase column (150 × 2.1 mm, Agilent particle size of 1.8 μM). The sample volume was 2 µL, and the mobile phase was divided into phase A and phase B (phase A: 0.1% formic acid in water, phase B: acetonitrile of chromatographic grade). The mobile phase gradient conditions were as follows: 95% A for 35 min, maintain 5% A for 10 min in the middle and then 95% A for 15 min. The flow rate was controlled at 0.3 mL/min, 40 °C. Typical parameters of the ESI source were as follows: ESI positive mode, spray voltage 3200 v, nitrogen as auxiliary gas pressure 34.48 kPa, nitrogen sheath pressure 206.89 kPa, capillary temperature, and ion source temperature 350 °C. Data Analysis software was applied for further data acquisition and processing.

### 2.6. Biochemical Characterization of Transaminases

The biochemical characterization of transaminases was identified by HPLC analysis as mentioned in Section 2.4. The degradation rate of HFB1 = (original content of HFB1−remaining content of HFB1)/original content of HFB1 × 100%. To investigate optimum temperature, the enzyme activity was detected at a temperature range of 30–70 °C. In order to determine thermostability, purified enzymes were incubated at 50 °C and 60 °C for different times without substrate, and the enzyme activities were measured under appropriate reaction conditions, respectively. The effects of pH on enzyme activity were determined at different pH values, the used buffers contained 0.1 mol/L citric acid—Na_2_HPO_4_ (pH 3.0–8.0), 0.1 mol/L Tris—HCl (pH 8.0–9.0), and 0.1 mol/L glycine—NaOH (pH 9.0–10.0). The pH stability of enzyme was assessed by incubating at various pH levels (3.0–10.0) for 1 h at 37 °C. The untreated enzyme was used as a control.

## 3. Results and Discussions

### 3.1. Gene Mining of FB1 Detoxifying Transaminases

Food and feed contaminated with mycotoxins worldwide pose a severe threat to the health of human and animal, resulting in significant economic losses annually [2]. Biotransformation by enzyme is one of the most attractive and potential alternatives to minimize the detrimental effects of mycotoxins [29,30]. Previously, transaminases were reported to effectively remove group C2-amino and convert FB into environmentally safe non-toxic or low-toxic products without destroying the nutritional content of food [23]. To date, only two transaminases FumIB (GenBank No. ADO15008.1) and FumIS (GenBank No. 6HBS_A) were reported to detoxificate FB1 [26,28]. Thus, more novel and effective fumonisin detoxification transaminases should be explored and characterized.

For the present study, FB1 detoxification transaminase candidates were validated by several pathways, as shown in Figure 1. Amino acid sequence analysis showed that FumIB and FumIS are very diverse, the sequence identity between them is only 65%. In order to obtain more potential FB1 detoxifying transaminases, FumIB and FumIS were used as model by BLASTp analysis, respectively. The results found that FumIB and FumIS both displayed low identity (70.77% and 75.59%) to the submitted sequences from Genbank database, which suggests the novelty of FB1 detoxifying transaminases. Then, the top five sequences of each transaminase with the highest identity were downloaded to construct an alternative gene pool. After removing the redundant sequences with identity >95%, three candidate proteins named FumTSTA, FumPHTA and FumUPTA were obtained (WP_089262923.1, PZQ64330.1, WP_068876442.1), which shared 64.80%, 63.72%, and 74.19% identity with FumIS, and 64.95%, 61.86%, and 66.28% identities with FumIB (Table 1). Moreover, the identity between each other was relatively low 25%, which indicates that three transaminases might be novel FB1 detoxifying transaminases.

Previous studies indicated that transaminases contain five families based on sequence similarity and protein structure [31]. Three transaminases FumTSTA, FumPHTA and FumUPTA were used to construct a rootless phylogenetic tree with reference transaminases from five classes (Figure 2), which showed that FumTSTA, FumPHTA, and FumUPTA, and two previously reported fumonisin detoxification transaminases (FumIB and FumIS) all belong to class III transaminases (Figure 2). Moreover, catalytic sites of class III transaminases were also found in sequence alignment. The catalytically active residue Lys located in the loop that links the β9 and β10 chains (Figure 3) [32].

Moreover, the three-dimensional structures of the five transaminases were modeled by Alphafold2 and simultaneously compared by Pymol (Figure 4), and they shared the similar tertiary protein structure (Figure 4). Moreover, the RMSD value is used to estimate the protein structure similarity and reveal the discrepancies in the structure of the two proteins. The smaller RMSD value is, the more similar the two protein structures are [33]. The RMSD values between three transaminases and two reported FumIB and FumIS are all below 0.37 (Table 1). These results suggested that the internal organization of these enzymes has a high degree of unity, indicating that its binding domain and channel mightly maintain structural consistency, allowing it to be employed as a new enzyme to mine new genes. Since three transaminases FumTSTA, FumPHTA and FumUPTA shared high similarity in protein structure and identical catalytic sites, they were chosen for the following expression and characterization experiment.

### 3.2. Protein Expression and Purification

The gene sequences of FumTSTA, FumPHTA, and FumUPTA were artificially synthesized and cloned into the expression vector pET-28a(+)and successfully transformed into *E. coli* BL21 (DE3). The recombinant proteins were further induced by IPTG addition, and the intracellular supernatant extracted by ultrasonic wall breaking were used to detect HFB degradation activity. After purification by Ni-affinity chromatography, purified FumTSTA, FumPHTA, and FumUPTA migrated as a single band on SDS-PAGE with a molecular weight of approximately 55 kDa (Figure 5A), which is consistent with the calculated molecular weight of recombinant protein. Western blot was also used to confirm the expression of the target gene in *E. coli*, which shows the consistent molecular weight of target recombinant protein (Figure 5B).

To evaluate the fermentation level, the effects of induction temperature, IPTG concentration, and incubation time on protein expression were relatively analyzed. As shown in Figure 6, the optimal induction conditions of three transaminases are diverse. The optimal temperature of FumTSTA, FumPHTA and FumUPTA is 20 °C, 25 °C, and 30 °C. The optimal IPTG concentrations of FumTSTA and FumPHTA are both 0.8 mM, and FumUPTA showed the highest yield at 0.5 mM. The optimal induction time is 20 h, and the protein concentration did not increase over time. Under the optimal induction condition, FumTSTA showed the highest yield of 94 mg/L, and the final protein yields of FumPHTA and FumUPTA were 39 mg/L and 34 mg/L, respectively. Previously, FumIB and FumIS were both expressed in *E*. *coli* [26,28]. However, FumIS tended to form inclusion bodies in *E*. *coli*, which severely hinders the soluble expression and practical application. The protein solubility of FumIS were further enhanced by exchanging host strain and co-expression of cold-adapted chaperonins [34]. The final concentration of recombinant proteins FumIS reached to 1.45 mg/L recombinant protein [28]. Thus, compared with FumIS, FumTSTA, FumPHTA and FumUPTA showed better protein solubility and higher protein yield, which suggests the advantage of FumTSTA, FumPHTA and FumUPTA.

### 3.3. LC-MS Analysis

The degradation vitalities of three transaminases against HFB1 were measured by HPLC. As shown in Figure 7, transaminases FumTSTA, FumPHTA, and FumUPTA can completely degrade HFB1, respectively, which suggests that all three transaminases have HFB1 detoxification activity. To further identify the degradation products of HFB1 by three transaminases, substrate HFB1 and degradation products were determined by LC-MS/MS, separately. As shown in Figure 8, the substrate HFB1 molecule of the protonated cation [M+H]^+^ was identified by QTOF-MS at m/z 406.3534 (Figure 8C), and the fragment ions such as m/z 316.3046, 334.3154, 352.3266, 370.3374 were also determined by secondary mass spectrometry (Figure 8D). These findings were consistent with the previously reported characteristics of HFB1 fragments [14].

The detoxification products of FumUPTA were in m/z 405 and m/z 448 (Figure 9 and Figure 10). In secondary mass spectrometry analysis of molecular ion m/z 405, fragment ions m/z 96.0814, 114.0927, 132.1035 were also detected (Figure 9D). The molecular formula of m/z 405 is inferred to be C_22_H_44_NO_6_ by the analysis software. Therefore, this molecular is presumably produced by the oxidation of the amino on HFB1. The secondary mass spectra of molecular ion m/z 448 were analyzed, the typical fragment ions of m/z 334.3155, 352.3268, 370.3374 and 388.3483 of 2-acetamide-HFB1 were also identified (Figure 10D). The molecular formula is inferred to be C_24_H_49_O_6_ by the analysis software. These results demonstrated that FumUPTA collaborate with pyruvate to acetylate or oxidize the amino functional group of HFB1, resulting in 2-acetamide-HFB1 and 2-keto-HFB1. Previous studies have shown that there are two kinds of products for transaminases FumIB [35]. The product of 2-keto-HFB1 is non-toxic. It is particularly non-toxic to GES-1 cells [23]. Different from FumUPTA, only the products m/z 448 were detected in the degraded products of FumTSTA and FumPHTA. It might imply that the catalytic mechanisms of these transaminases are different, although they all belong to Class III transaminases.

### 3.4. Degradation Activity and Biochemical Characteristics

The biochemical characteristics of three transaminases were comparatively analyzed. The optimum temperature of these transaminases is 35 °C and 40 °C (Figure 11A–C), which is not only close to the physiological temperature in the GIT of pigs and cattle but in all homeothermic vertebrates and enables them to perform their functions well in feed or animals [29]. Moreover, three transaminases displayed excellent activity at high temperatures: FumTSTA showed 70% relative activity at 50 °C, FumPHTA maintained more than 40% relative activity in the temperature range from 50 °C to 60 °C (Figure 11A,B), FumUPTA performed more than 40% relative activity in the range of 50–70 °C (Figure 11C). The broad reaction temperature range largely broadens the application aspects of fumonisin detoxification transaminases.

Thermostable enzymes are also extremely critical for industrial field due to their specific processes and low cost [36]. FumTSTA, FumPHTA, and FumUPTA were able to maintain 73%, 72%, and 91% relative activity after incubated at 60 °C for 5 min (Figure 12A–C), 27%, 28% and 18% relative activity after 1 h treatments at 60 °C (Figure 12A–C). Transaminases FumTSTA and FumPHTA showed similar thermostability. FumUPTA displayed better thermostability than previous two transaminases after incubated at 60 °C for 5 min, but worse thermostability after incubated at 60 °C for 1 h. Moreover, based on the thermostability of FumTSTA, FumPHTA, and FumUPTA might not endure the pelleting heating process, but using an adequate coating of these enzymes to avoid their thermal inactivation is also possible and possible way. The stability could be further improved by molecular modification.

As shown in Figure 11, three transaminases displayed different optimal pH range. FumTSTA was most active at pH 9.0 and retained activity in acidic and neutral conditions (Figure 11D). Compared with FumTSTA, FumPHTA had a lower optimal reaction pH of 8.0 and showed high relative activity (>59.89%) at a neutral pH range of 6.0–8.0 (Figure 11E). Previous studies showed that FumIS is also alkaline with pH optimum of 8.5, but its pH active range is relatively narrow, from pH 7.0 to pH 9.0 [28]. Notably, different from three transaminases, FumUPTA displayed its optimal reaction pH of 4.0, and it exhibited more than 50.0% relative activity from pH 3.0 to 5.0 (Figure 11F). Additionally, FumTSTA exhibited good pH stability in a wide pH range from pH 3.0 to pH 10.0 (Figure 12D). However, the pH stability of FumPHTA is not as good as that of FumTSTA, which only retains 40% residual activity at pH 3.0 and 4.0 (Figure 12E). In contrast, FumUPTA is active in acidic condition and also showed excellent pH stability under acidic conditions (pH 3.0–6.0), which retains 60% residual activity from pH 3.0–6.0 for 1 h (Figure 12F).

Fumonisin has widely existed in many application scenarios, including different food and feed materials. The pH condition of actual application scenarios for fumonisin detox enzymes are divergent. For example, the physiological pHs of the animal intestinal environment are acidic typically between 5.0 and 6.5 [29], while the gastric pHs of certain monogastric animals, such as pigs, are around 2.0 [37]. Moreover, the corn steep water for amylase production is acidic around pH 3–4 [38]. Furthermore, fumonisins have been extensively detected in manufactured aquatic feed, while the pH of fish intestinal tract in aquaculture is neutral [39,40]. Thus, fumonisins detoxification enzymes with different pH are required for removing FB in practical applications. One enzyme usually displays a narrow pH optimum. Herein, the divergent pH optimum of three transaminases significantly enlarges the application scenes of fumonisin degradation transaminases and provide more enzyme candidates, which can be used individually or combinatorially.

## 4. Conclusions

In this study, three novel transaminases were identified to effectively detoxify hydrolyzed fumonisin B1 (HFB1) by gene mining and enzymatic characterization. They displayed diverse pH reaction ranges, good pH stability and thermostability, and high protein yield, which provides ideal candidates for fumonisin detoxification and meets different required demands in food and feed industries.

## Figures and Tables

**Figure 1 foods-12-00416-f001:**
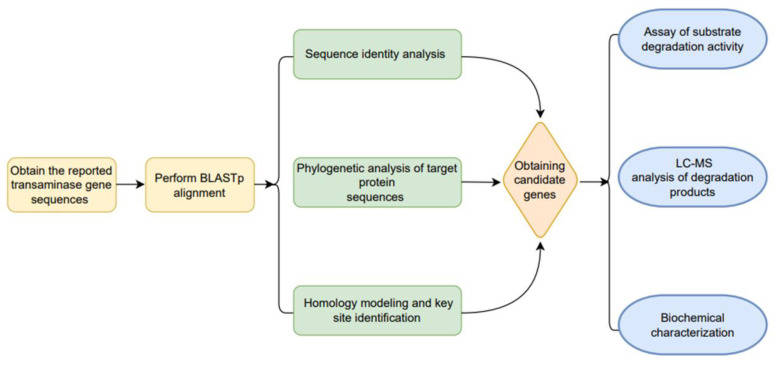
Screening and identification of FB1 detoxification transaminases.

**Figure 2 foods-12-00416-f002:**
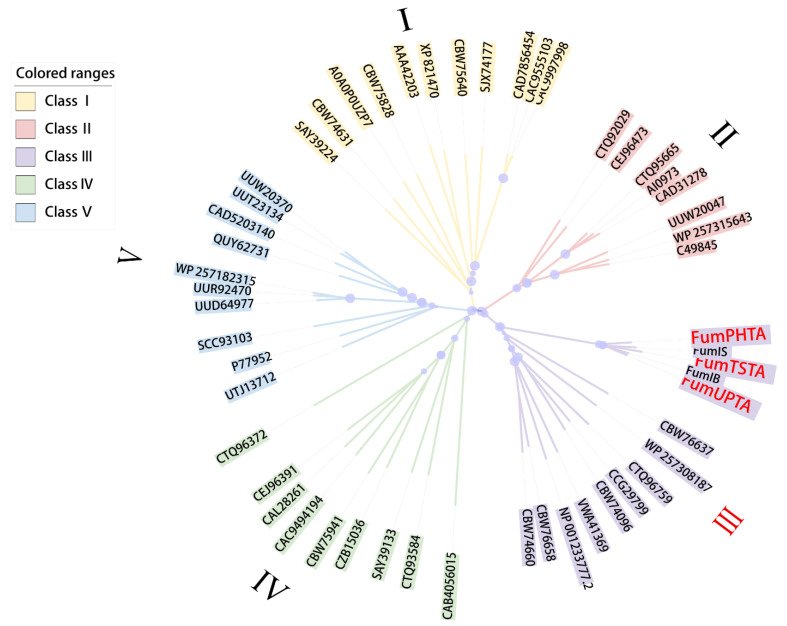
Establishment of a phylogenetic tree of fumonisin detoxification transaminases and other fold type transaminases sequences from five class. Branch lengths reflected the relative variation in amino acid sequences from each other. The light blue circle symbols were the bootstrap values of 1000 repetitions, the circle size was directly related to the number of bootstrap values. The corresponding family was shown. Three transaminases are highly marked in red letters.

**Figure 3 foods-12-00416-f003:**
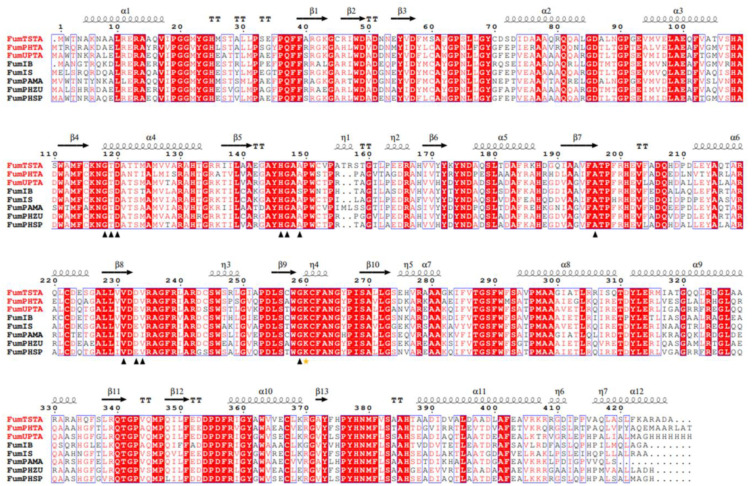
The amino acid sequence alignment regarding three transaminases with the other homologous sequences. These transaminases were from *Paraburkholderia madseniana* (WP_154567590), *Phenylobacterium* sp (TAJ71837.1, PZQ64330.1). Two reported two different fumonisin transaminases FumIS from *Sphingopyxis* sp. MTA144 (6HBS_A), FumIB from bacterium ATCC 55552 (ADO15008.1). The identified key active site residues in the small (S) pocket are indicated by black triangles. The catalytically active Lys residue are indicated by an orange pentagram. The strictly conserved residues are indicated in red, and the highly conserved residues are indicated in red.

**Figure 4 foods-12-00416-f004:**
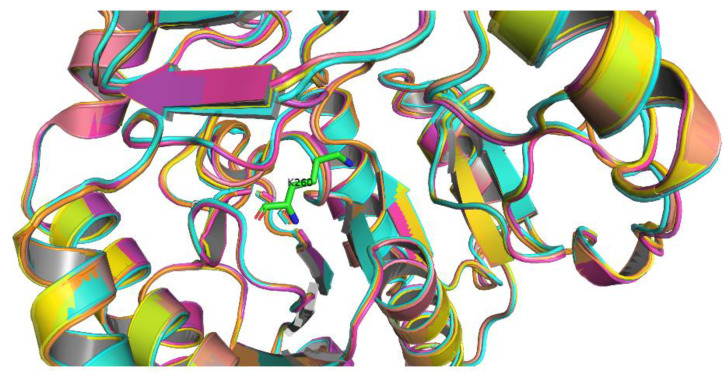
Superimposition of the crystal structure from FumIS (salmon), FumIB (yellow), FumTSTA (orange), FumPHTA (cyans), and FumUPTA (magenta) transaminases; Green stick structure represent the catalytically active Lys residue of FumTSTA.

**Figure 5 foods-12-00416-f005:**
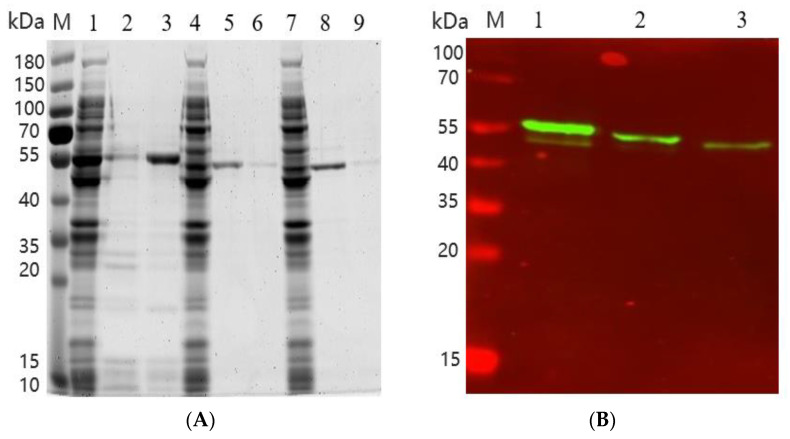
(**A**) SDS-PAGE of the recombinant protein. Lane M, molecular weight marker. Lane 1, crude enzyme solution of FumTSTA. Lane 2, purified recombinant FumTSTA (200 mM Imidazole elution). Lane 3, purified recombinant FumTSTA (300 mM Imidazole elution). Lane 4, crude enzyme solution of FumPHTA. Lane 5, purified recombinant FumPHTA (200 mM imidazole elution). Lane 6, purified recombinant FumPHTA (300 mM Imidazole elution). Lane 7, crude enzyme solution of FumUPTA. Lane 8, purified recombinant FumUPTA (200 mM imidazole elution). Lane 9, purified recombinant FumUPTA (300 mM Imidazole elution); (**B**) Western-blotting validation of transaminase protein, M: Marker. Lane 1, FumTSTA. Lane 2, FumPHTA. Lane 3, FumUPTA.

**Figure 6 foods-12-00416-f006:**
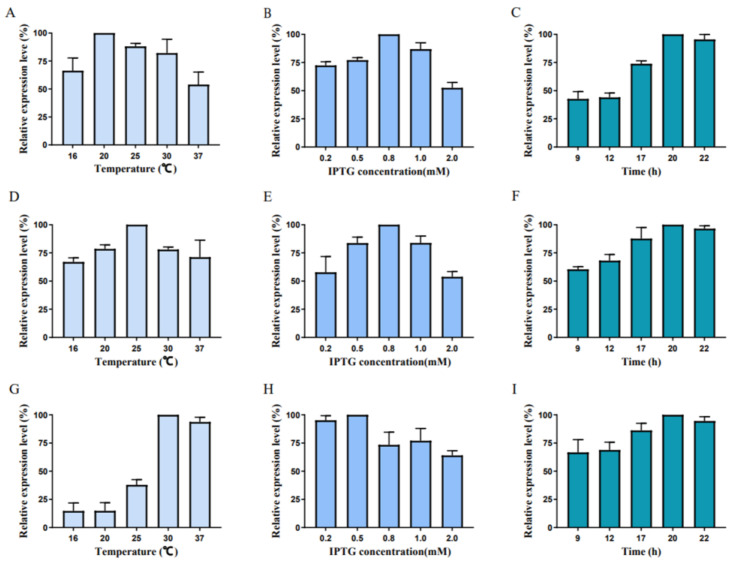
Fermentation level optimization of recombinant proteins. (**A**) Different induction temperatures of FumTSTA; (**B**) Different IPTG concentration of FumTSTA; (**C**) Different incubation time of FumTSTA; (**D**) Different induction temperatures of FumPHTA; (**E**) Different IPTG concentration of FumPHTA; (**F**) Different incubation time of FumPHTA; (**G**) Different induction temperatures of FumUPTA; (**H**) Different IPTG concentration of FumUPTA; (**I**) Different incubation time of FumUPTA.

**Figure 7 foods-12-00416-f007:**
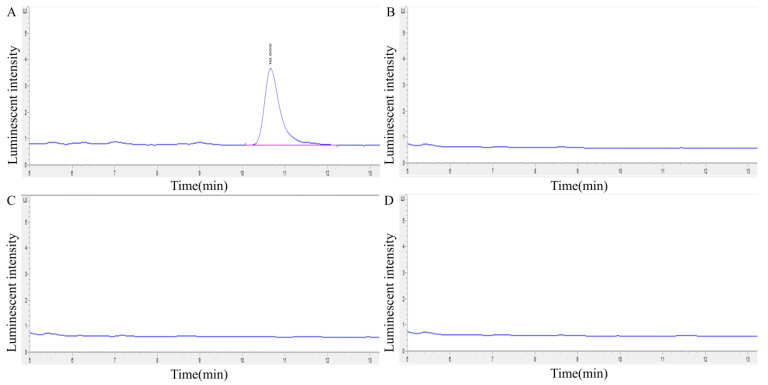
Analysis of the detoxification activity of transaminases by HPLC. (**A**) Chromatogram of HFB1; (**B**) Chromatogram of FumTSTA degradation of HFB1; (**C**) Chromatogram of FumPHTA degradation of HFB1; (**D**) Chromatogram of FumUPTA degradation of HFB1.

**Figure 8 foods-12-00416-f008:**
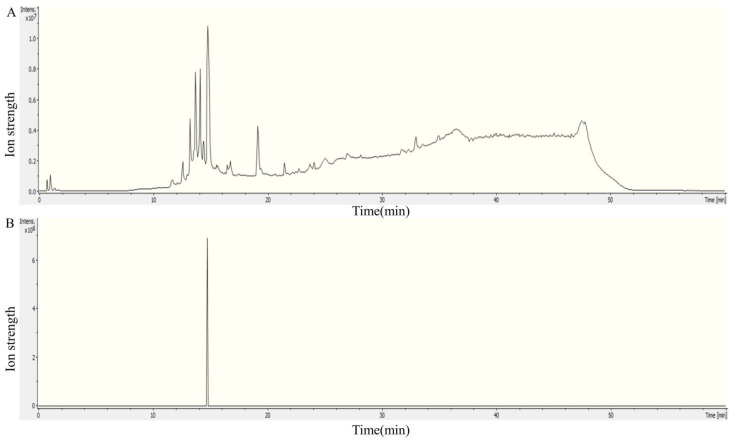
Mass spectrum of HFB1. (**A**) Total ion flow diagram; (**B**) The EIC chart of HFB1; (**C**) The primary mass spectrum of HFB1; (**D**) The secondary mass spectrum of HFB1.

**Figure 9 foods-12-00416-f009:**
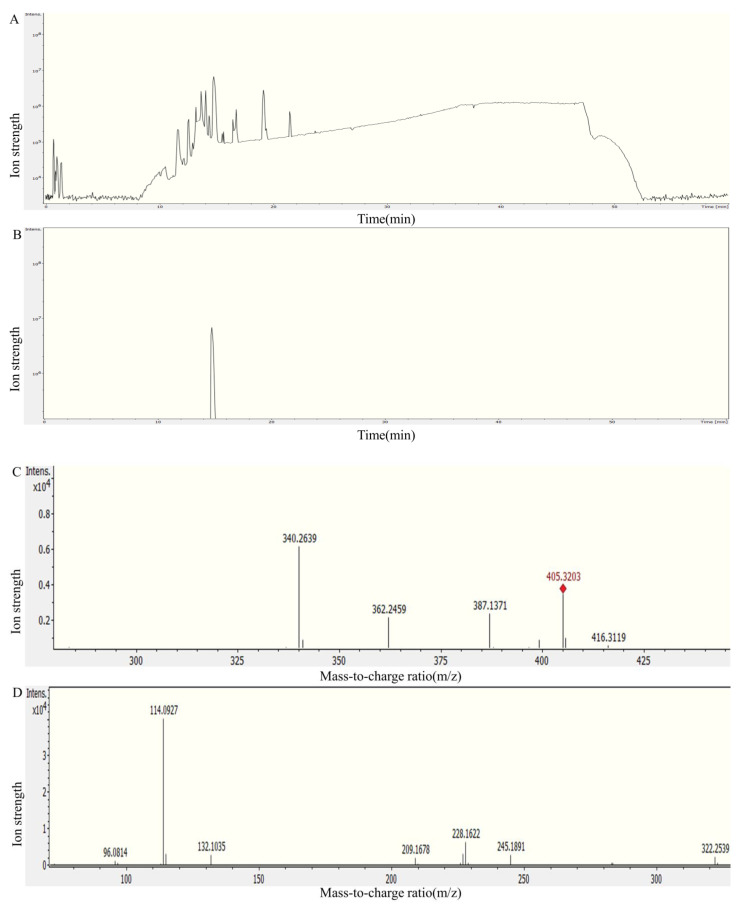
Analysis of FumUPTA degradation of HFB1 by mass spectrometry. (**A**) Total ion flow diagram; (**B**) The EIC chart of 2-keto-HFB1; (**C**) The primary mass spectrum of 2-keto-HFB1; (**D**) The secondary mass spectrum of 2-keto-HFB1.

**Figure 10 foods-12-00416-f010:**
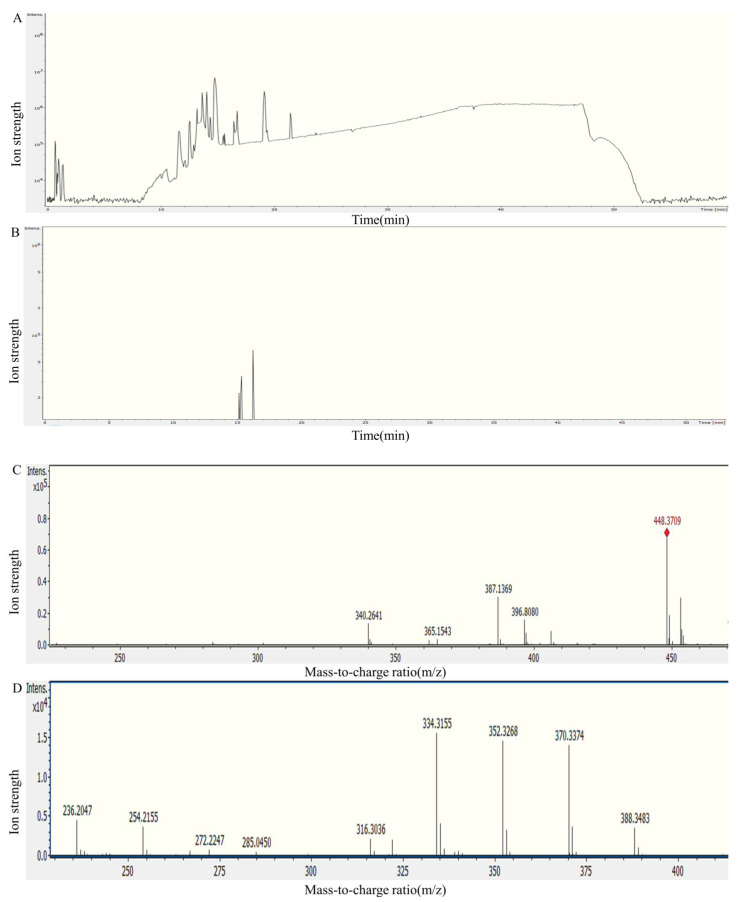
Analysis of FumUPTA degradation of HFB1 by mass spectrometry. (**A**) Total ion flow diagram; (**B**) The EIC chart of 2-acetamide-HFB1; (**C**) The primary mass spectrum of 2-acetamide-HFB1; (**D**) The secondary mass spectrum of 2-acetamide-HFB1.

**Figure 11 foods-12-00416-f011:**
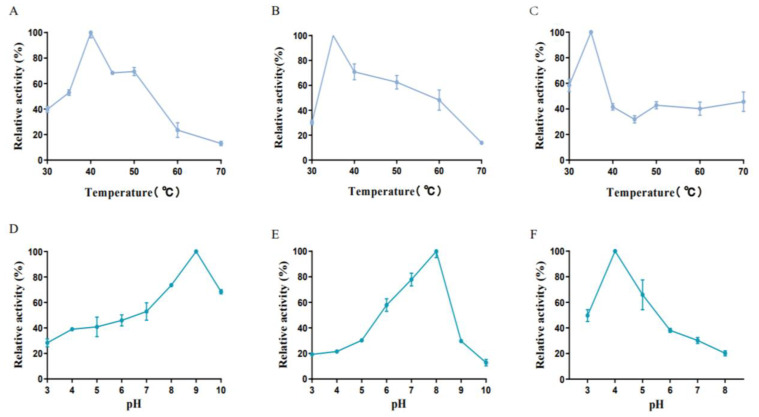
Characterization of purified recombinant enzymes. (**A**) Effect of temperature on FumTSTA catalytic activity; (**B**) Effect of temperature on FumPHTA catalytic activity; (**C**) Effect of temperature on FumUPTA; (**D**) Effect of pH on FumTSTA catalytic activity; (**E**) Effect of pH on FumPHTA catalytic activity; (**F**) Effect of pH on FumUPTA catalytic activity.

**Figure 12 foods-12-00416-f012:**
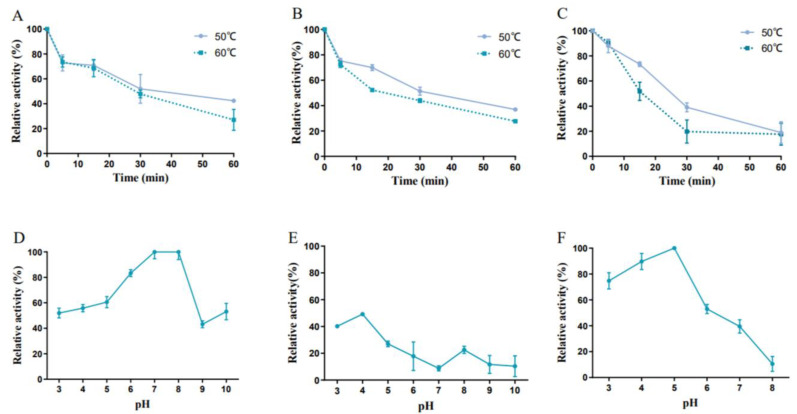
Characterization of purified recombinant enzymes. (**A**) Thermostability of FumTSTA; (**B**) Thermostability of FumPHTA; (**C**) Thermostability of FumUPTA; (**D**) pH stability of FumTSTA; (**E**) pH stability of FumPHTA; (**F**) pH stability of FumUPTA.

**Table 1 foods-12-00416-t001:** Analysis of sequence identity and RMSD value of transaminases.

	Sequence Identity (%)		RMSD Value	
	FumIB	FumIS	FumIB	FumIS
FumTSTA	64.95	64.80	0.027	0.058
FumPHTA	61.86	63.72	0.021	0.058
FumUPTA	66.28	74.19	0.022	0.051

## Data Availability

The data presented in this study are available within the article.

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
