# Peer review of "Detoxification of Fumonisins by Three Novel Transaminases with Diverse Enzymatic Characteristics Coupled with Carboxylesterase"

_foods, 2023, doi:10.3390/foods12020416_

Round 1
Reviewer 1 Report
Congratulations to the authors for a well-designed and performed study on fumonisin detoxification. Three fumonisin detoxification transaminases with were identified and studied, which have excellent catalytic properties, according to the authors. Additionally, they have diverse enzymatic characteristics compared with other reported transaminases (broader pH reaction range and better pH stability and thermostability), which are excellent characteristics from the aspect of chemical and mechanical stability.
Methods are adequately described, results are clearly presented, and conclusions are supported by the results. The introduction provides sufficient background, and references are appropriate. I do not have further comments on improving the manuscript.
Author Response
Thanks very much for your reviewing.
Reviewer 2 Report
The manuscript "Detoxification of fumonisins by three novel transaminases 2 with diverse enzymatic characteristics coupled with carboxyles-3 terase" by Wang et al. addresses the important issue of fumonisin detoxification. This is an important topic that requires attention. The manuscript has a number of important weaknesses. The authors have not read or cited the published literature on this topic. There are a number of papers and products on the market that already address fumonisin detoxification. The transaminases were identified by blast searching known ones, this is fine but the authors should have compared the known one in their experiments for a direct control. Without these data it is impossible to know how well these new enzymes work in comparison to those previously reported. The LC-MS data are poorly presented as screenshots, this is not acceptable. The authors suggest that these could be used for food, there is no evidence anywhere that they would be approved for food for human consumption. Overall the English is weak and the manuscript although interesting, does not provide very much new info.
Author Response
Response to Reviewer 2 Comments
The manuscript "Detoxification of fumonisins by three novel transaminases 2 with diverse enzymatic characteristics coupled with carboxyles-3 eterase" by Wang et al. addresses the important issue of fumonisin detoxification. This is an important topic that requires attention. The manuscript has a number of important weaknesses. The authors have not read or cited the published literature on this topic. There are a number of papers and products on the market that already address fumonisin detoxification.
Response: Thanks very much for your suggestion. We agree that a lot of research work and products for fumonisin detoxication, and the researches were added in the manuscript. To our best knowledge, there are many approaches for eliminating fumonisin, and enzymatic is one of them. However, only limited transaminases were reported to degrade fumonisin, thus this study focus on exploring the transaminases for fumonisin detoxication.
Point 1: The transaminases were identified by blast searching known ones, this is fine but the authors should have compared the known one in their experiments for a direct control. Without these data it is impossible to know how well these new enzymes work in comparison to those previously reported.
Response 1: We agree with reviewer’s suggestion. It is better to compare these three transaminases with previous reported FumIS from Sphingopyxis sp. MTA144 and FumIB from bacterium ATCC 55552 in the same experiment. While, it might be also acceptable for comparision of optimal temperature, pH based on the literature. And some sentences in this manuscript were also modified. Perhaps, this data could be added in future reasearch. Thanks for your understanding.
Point 2: The LC-MS data are poorly presented as screenshots, this is not acceptable.
Response 2: We appreciate it very much for this good suggestion, and we have changed the LC-MS data with higher dpi value in Figure 8-10.
Point 3:The authors suggest that these could be used for food, there is no evidence anywhere that they would be approved for food for human consumption.
Response 3: Thanks for the reviewer's suggestion. This study provides ideal transaminases candidates for fumonisin detoxification, which have greater potential for food and feed applications in the future. Indeed, there are a lot of work needed to do before they are used in food and feed industries, thus the sentences were revised in the manuscirpt. Line 70-74.

Reviewer 3 Report
I have carefully read the manuscript by Wang et al and in my opinion the experimental section is well planned. However, I suggest a careful revision before possible consideration for publication as reported below.
Page 3, lines 142-144: please revise the sentence. It is unclear. Which volume was used for the assay? Which amount of FB1 10 ug/mL used? The authors used 100 mL of enzyme?? Which amount? Which concentration of the enzyme? Please clarify.
Page 4, line 155: The authors indicate just the volume of enzyme used. What is the stock concentration? Which amount is used in the assay? Please indicate concentration (e.g. µg/µL) for each enzyme assayed.
Page 4, paragraph 2.6. Please specify the assay used to evaluate the enzymatic activity of transaminases. Describe the method.
Page 9, Figure 6. Please revise the Figure showing all error bars. I suggest changing the maximum in ordinate axis.
Figures 8, 9 and 10: The Figure are not clear. Please increase the resolution
Figure 11: As for Figure 6, please revise the error bars and increase the resolution of the graphs.
Page 12, lines 380-376: The sentence is not clear. The authors reported a difference in residual activity after 5 min and 1 h, without discussing the relative graph. Moreover, in Figure 11 A, the residual activity is of about 20%, while in Figure 11C is about 50%. However, in the text, the authors reported different results. Please clarify.
Please the reference format, according to Foods reference style.

Author Response
Response to Reviewer 3 Comments
Point 1: Page 3, lines 142-144: please revise the sentence. It is unclear. Which volume was used for the assay? Which amount of FB1 10 ug/mL used? The authors used 100 mL of enzyme? Which amount? Which concentration of the enzyme? Please clarify.
Response 1: Thanks for reviewer’s comments. We have revised the content in line 130-132.
Point 2: The authors indicate just the volume of enzyme used. What is the stock concentration? Which amount is used in the assay? Please indicate concentration (e.g. µg/µL) for each enzyme assayed.
Response 2: We appreciate it very much for this good suggestion, and we have revised the content in line 134-137.
Point 3: Please specify the assay used to evaluate the enzymatic activity of transaminases. Describe the method.
Response 3: The sections were further specified. Line 165-167.
Point 4: Figure 6. Please revise the Figure showing all error bars. I suggest changing the maximum in ordinate axis.
Response 4: Thanks for reviewer’s comments. The ordinate axis represent the relative values. In each figure, the highest value are set as 100% (control), thus there is no significant error bars for them, such as the 20℃in Figure 6A. The figure will be simialr when enlarging the the maximum in ordinate axis, maybe 100% is OK for the the maximum in ordinate axis.
Point 5: Figures 8, 9 and 10: The Figure are not clear. Please increase the resolution.
Response 5: Thanks for reviewer’s comments. We have modified the image in Figure 8,9 and 10.
Point 6: Figure 11: As for Figure 6, please revise the error bars and increase the resolution of the graphs.
Response 6: Thanks for reviewer’s comments. We have modified the image in Figure 11.
Point 7: Page 12, lines 380-376: The sentence is not clear. The authors reported a difference in residual activity after 5 min and 1 h, without discussing the relative graph. Moreover, in Figure 11 A, the residual activity is of about 20%, while in Figure 11C is about 50%. However, in the text, the authors reported different results. Please clarify.
Response 7: These sentences were rewritten as reviewer’s suggestion. Line 369-374.
Please the reference format, according to Foods reference style.
Response 8: The reference format were all carefully checked and revised.

Reviewer 4 Report
Review for
Detoxification of fumonisins by three novel transaminases with diverse enzymatic characteristics coupled with carboxylesterase
Detoxification of fumonisins is a topic under investigation by many teams
Protocatechuic acid: A novel detoxication agent of fumonisin B1 for poultry industry
Wang, F., Chen, Y., Hu, H., (...), He, C., Haque, M.A. 2022 Frontiers in Veterinary Science
9,923238
Degradation of four major mycotoxins by eight manganese peroxidases in presence of a dicarboxylic acid
Wang, X., Qin, X., Hao, Z., (...), Yao, B., Su, X. 2019 Toxins
11(10),566
Microbiologicals for deactivating mycotoxins Schatzmayr, G., Zehner, F., Täubel, M., (...), Loibner, A.P., Binder, E.M. 2006 Molecular Nutrition and Food Research
50(6), pp. 543-551
Fumonisin (FB) is one of the most common mycotoxins contaminating feed and food, causing severe public health threat to human and animals worldwide. Until now, only several transaminases were found to reduce FB toxicity, thus, more fumonisin detoxification transaminases with excellent catalytic properties were urgent to be explored for complex application conditions.
In the present study, through gene mining and enzymatic characterization, three novel fumonisin detoxification trans-aminases FumTSTA, FumUPTA, FumPHTA were identified by the authors, sharing only 61-74% sequence identity with reported fumonisin detoxification transaminases.
Besides, the recombinant proteins shared diverse enzymatic characteristics compared with previous reported transaminases, such as broader pH reaction range and better pH stability and thermostability.
The study was well designed, and conducted.
The results support the interpretation done.
Discussion is helpful for the readers.
One question: how these transaminases could be applied at ton-scale?
In so many animal feed and human food (solid, liquid)?
Author Response
Response to Reviewer 4 Comments
Point 1: how these transaminases could be applied at ton-scale? In so many animals feed and human food (solid, liquid)?
Response 1: Thanks for reviewer’s comments. This study provides ideal transaminases candidates for fumonisin detoxification, which have greater potential for food and feed applications. Generally, transaminases could be applied at ton-scale in different application scenarios. Firstly, transsaminase can be largely produced by fermentation and separation process as other enzyme additives. Then, the practical application method depends on the different application scenarios, like enzyme power (solid) could be added and used in raw material to produce fermented feed, and liquid enzyme will be better for fumonisin-contaminated corn steep liquor to remove fumonisin. Apparently, there are still a lot of work to do for transaminases application, like the catalyitc actvity could be further improved and the protein yield should be increased in future to reduce the production cost.

Reviewer 5 Report
The paper's topic is to use three novel transaminases by three novel transaminases coupled with carboxylesterase for the detoxification of fumonisin B1. It would be an important tool for the food and feed industry to decrease the toxicity of FB1 in animals and humans.
The paper is well structured and written and contains a detailed dataset about the biochemical characteristics of the three novel transaminases.
The introduction is correct and contains relevant references to previous studies in that field.
Materials and Methods are also correct, but some information is lacking.
Results and Discussion is informative and shows all relevant information. However, additional experimental data or proposals for additional studies would be helpful for further industrial application.
No information about the purity and manufacturer of FB1 was used in the degradation study.
There is no information about the obtaining of HFB1 after HPLC detection. It is unclear whether the HFB1 obtained from the previous step by preparative HPLC or purified HFB1 was used for the second enzymatic degradation step.
Please, specify the purified transaminase solution (e.g., protein content). There are some data about the protein yield of the enzymes (Results and Discussion 2.3.), but not about the purified solution used in the enzymatic degradation study.
It would be useful to determine the thermostability of the enzymes over 60 oC because the temperature is higher during food and feed processing (e.g., sterilisation or pelleting).
Please, add a relevant reference to the non-toxic properties of 2-acetamide-HFB1.
L 370-372: The temperature range of 35-40 oC is close not only to the physiological temperature in the GIT of pigs and cattle but in all homeothermic vertebrates, including humans.
L 380-384: Please, refer here that based on the results, FumUPTA has high thermostability and can be proposed for industrial application. However, the other possible way is to use an adequate coating of the final product to avoid the thermal inactivation of the enzymes.
L 394-408: The pH stability of the three enzymes is different, but it would be important to know the pH stability in the stomach of some monogastric animals, such as pigs, which is around 2.0. It would be useful to know the "recovery" of the enzyme activity first at low and after moderately low pH conditions, which happened in the GIT.
Author Response
Response to Reviewer 5 Comments
Point 1: No information about the purity and manufacturer of FB1 was used in the degradation study.
Response 1: Thanks for reviewer’s comments. The purity and manufacturer of FB1 were added in section 2.1. “Substrate FB1 (purity of 98%) was obtained from Pribolab Ltd. (Qingdao, China, http://wwwpribolab.com/) and dissolved in acetonitrile-water.” line 86-88.
Point 2: There is no information about the obtaining of HFB1 after HPLC detection. It is unclear whether the HFB1 obtained from the previous step by preparative HPLC or purified HFB1 was used for the second enzymatic degradation step.
Response 2: We appreciate it very much for this good suggestion. These informantion were added as follow. “In this study, HFB1 was firstly obtained from the reaction mixture of FB1 and esterase FumDSB, 100 µl of FB1 solution (50 µg/mL) and 900 µl of purified enzyme FumDSB (1.164 mg/mL) were reacted at 37 °C for 2 h. The producing HFB1 were further identified by LC-MS, then they were used as the substrate for transaminases directly without any extraction and purification steps.” line 130-134.
Point 3: Please, specify the purified transaminase solution (e.g., protein content). There are some data about the protein yield of the enzymes (Results and Discussion 2.3.), but not about the purified solution used in the enzymatic degradation study.
Response 3: the informantion of purified solution were added. “For transaminase, each assay solution (100 µL) contained 95 µL of purified transaminase (1.38 mg/mL) in PBS buffer (8 g/L NaCl, 3.58 g/L Na2HPO4·12H2O, 0.2 g/L KCl, 0.27 g/L KH2PO4), 5 µL of HFB1 solution, and pyruvate solution, mixed to a final concentration of HFB1 of 1 μg/mL.…” line 134-136.
Point 4: It would be useful to determine the thermostability of the enzymes over 60℃ because the temperature is higher during food and feed processing (e.g., sterilisation or pelleting).
Response 4: We agree with reviewer’s suggestion. As shown in Figure 12, the thermostability were not very well at 60℃. We also tried to test the thermostability at 70℃ for one time, they almost lost activity. However, because of the epidemic, this experiment can not be repeated for three times now and shown in manuscript. Due to the high efficency of transaminases, they show potential in industries. Thus,we also plan to further improve the thermostability in future. Thanks very much for your understanding.
Point 5: Please, add a relevant reference to the non-toxic properties of 2-acetamide-HFB1.
Response 5: Thanks for reviewer’s comments, to our best knowledge, there are no specific literature reports on whether it is toxic or not, because many work need to do. We will follow up.
Point 6: L370-372: The temperature range of 35-40 oC is close not only to the physiological temperature in the GIT of pigs and cattle but in all homeothermic vertebrates, including humans.
Response 6: This was added in the manuscript. “The biochemical characteristics of three transaminases were comparatively analyzed. The optimum temperature of these transaminases is 35 ℃ and 40 ℃ (Figure 11A, B and C), which is close to not only to the physiological temperature in the GIT of pigs and cattle but in all homeothermic vertebrates,…” line 357-360.
Point 7: L 380-384: Please, refer here that based on the results, FumUPTA has high thermostability and can be proposed for industrial application. However, the other possible way is to use an adequate coating of the final product to avoid the thermal inactivation of the enzymes.
Response 7: We appreciate it very much for this good suggestion. This is added into this section.
“Besides, based on the thermostability of FumTSTA, FumPHTA, and FumUPTA might not endure the pelleting heating process, but using an adequate coating of these enzymes to avoid their thermal inactivation is also possible and possible way. The stability could be further improved by molecular modification.” Line 376-379.
Point 8: The pH stability of the three enzymes is different, but it would be important to know the pH stability in the stomach of some monogastric animals, such as pigs, which is around 2.0. It would be useful to know the "recovery" of the enzyme activity first at low and after moderately low pH conditions, which happened in the GIT.
Response 8: We agree with the reviewer’s suggestion, this information is added in the article in line 404-406. Besides, the recovery of the enzyme activity is very a useful reference for evaluating the enzymes. However, this data are very hard to finish in this serious time, our lab is closed now because of the epidemic. This data will be furthr researched in our following applied research. Thanks very much for your understanding.

Round 2
Reviewer 2 Report
The authors have addressed a number of reviewer concerns. The big issue of not directly comparing the known enzyme for which these sequences were blasted from is a fatal experimental design flaw. The argument that the enzymes in this manuscript perform better is not valid as these experiments would need to be conduced side by side using enzymes that were prepared using the same methods and tested with the same purity of toxin. Unfortunately, more experiments are needed before this can be accepted. The chromatograms are also still poorly presented.
